# "It's behaviors, not identity": Attitudes and beliefs related to HIV risk and pre-exposure prophylaxis among transgender women in the Southeastern United States

Olivia T. Van Gerwen[1]*, Erika L. Austin[2], Andres F. Camino[3], L. Victoria Odom[3], Christina A. Muzny[1]

1 Division of Infectious Diseases, Department of Medicine, University of Alabama at Birmingham, Birmingham, Alabama, United States of America, 2 Department of Biostatistics, University of Alabama at Birmingham School of Public Health, Birmingham, Alabama, United States of America, 3 University of Alabama School of Medicine, Birmingham, Alabama, United States of America

* oliviavangerwen@uabmc.edu

**Data Availability Statement:** All relevant data are within the manuscript and its Supporting information files.

## Abstract

HIV prevalence is high among transgender women (TGW) in the Southeastern U.S. Uptake of HIV Pre-Exposure Prophylaxis (PrEP) is low among TGW nationwide. We aimed to explore beliefs associated with PrEP among TGW in the Southeastern U.S., framed by the Health Belief Model. HIV-negative TGW ≥18 years old in Alabama participated in virtual focus group discussions. Authors coded and amended transcripts to explore emerging themes. Between July-December 2020, 17 TGW participated in 4 sessions. Mean age was 28.1±8.5 years. Several themes were identified: frustration with conflation of transgender identity and HIV risk, inappropriate transgender representation in PrEP advertising, concerns for interactions between PrEP and hormone therapy, perception that PrEP is meant for cisgender men who have sex with men and limited trans-affirming healthcare. Nuanced messaging is necessary to properly educate and engage TGW in HIV prevention strategies including PrEP given the diversity of this population.

## Introduction

In the last decade, immense progress has been made toward ending the HIV epidemic, particularly in the United States [1]. This has been accomplished through treatment of those living with HIV as well as prevention strategies such as HIV Pre-Exposure Prophylaxis (PrEP) [2, 3]. Several clinical trials have demonstrated daily oral PrEP with the antiretroviral combinations of tenofovir and emtricitabine (TDF/FTC and TAF/FTC) to be well tolerated and effective in decreasing the risk of HIV acquisition in high-risk populations, such as cisgender men who have sex with men (MSM) and transgender women (TGW) [4–7].

Despite innovations such as PrEP, TGW continue to acquire HIV at astronomical rates [8]. Between 2009 and 2014, 2,351 transgender people in the U.S. were diagnosed with HIV, 84%

**Funding:** This work was supported by the UAB Center for AIDS Research [P30 AI027767-32 to OTVG] and the Agency of Healthcare Research and Quality [T32HS013852 to OTVG]. The funders had no role in study design, data collection and analysis, decision to publish, or preparation of the manuscript.

**Competing interests:** I have read the journal's policy and the authors of this manuscript have the following competing interests: OTVG has received research grant support from Gilead Sciences, Inc. and Abbott Molecular. CAM has received research grant support from NIH/NIAID, Gilead Sciences, Inc., and Lupin Pharmaceuticals, Inc.; serves as a consultant for Abbott Molecular, Lupin Pharmaceuticals, Inc., Cepheid, and BioFire Diagnostics; receives honoraria from Elsevier, Abbott Molecular, Cepheid, Becton Dickinson, Roche Diagnostics, and Lupin Pharmaceuticals, Inc. All other authors have no relevant disclosures. This does not alter our adherence to PLOS ONE policies on sharing data and materials.

of which were TGW [9]. TGW of color bear an even higher burden of HIV than other racial groups, with one meta-analysis estimating HIV prevalence to be 44.5% among black TGW, compared to 25.8% and 6.7% among Hispanic and white TGW, respectively [10]. Between 2009 and 2014, 43% of TGW diagnosed with HIV lived in the Southeastern U.S.; this region accounts for the highest number of new HIV infections annually [9, 11]. In addition, a 2016 national survey reported that the transgender population in the Southeastern U.S. is large and growing, with the number of individuals identifying as transgender in the state of Alabama, for example, exceeding 22,500 [12]. The disproportionate impact of HIV in the Southeastern U.S. coupled with the increasingly visible transgender population makes utilization of PrEP services by this marginalized community essential in achieving health equity and combating the HIV epidemic through prevention [13].

Despite the high rates of HIV among TGW in the Southeastern U.S., uptake of PrEP among this population remains low [14–16]. One recent multi-site U.S. cohort study of 600 TGW identified 47% to be eligible for PrEP, but only 28% of those reported using PrEP in the last 30 days [17]. Use of PrEP among eligible MSM, by comparison, was estimated to be 35% as of 2017 in another U.S. study [18]. Importantly, PrEP uptake varies widely regionally, with it being particularly low among MSM of color in the Southeastern U.S [19, 20]. While PrEP utilization among TGW in other regions of the U.S. (e.g., West Coast, Northeast) is more robust than in the Southeastern U.S. [14], this prevention strategy is underutilized by TGW nationwide when compared to other target populations such as white MSM [14]. A key gap in optimizing HIV prevention in Southeastern TGW is elucidating community-specific reasons for PrEP underutilization. One Southeastern study composed of 264 TGW and MSM recruited at a Gay Pride Festival found moderate PrEP awareness (63%) but very low uptake (9%), suggesting barriers at multiple points on the PrEP care continuum [21]. This study found stigma related to promiscuity and medical mistrust to be associated with lack of interest in PrEP [21]. However, this analysis did not specifically address the unique perspectives and barriers faced by the transgender community since it combined data for MSM and TGW [22]. Qualitative data from San Francisco, CA suggest that integration of the unique needs and characteristics of the transgender community into PrEP promotion strategies is essential to improving PrEP uptake in this population [15]. The intersection of regional, cultural, and gender-related challenges likely play a role in the lack of awareness and uptake of PrEP in Southeastern TGW. Given the critical gap in knowledge on this subject, this qualitative study aimed to identify barriers and beliefs experienced by TGW in the Southeastern U.S. regarding knowledge of and access to PrEP services for HIV prevention through virtual focus groups (VFG).

## Methods

### Study sample and recruitment methods

We recruited VFG participants using the following inclusion criteria: self-identification as a TGW, age ≥18 years, self-reported HIV-negative status, and residence in the state of Alabama. Exclusion criteria were significant cognitive impairment, active psychotic disorder, and non-English speaking. Additionally, since focus groups were conducted virtually due to the COVID-19 pandemic, participants had to have access to a reliable wireless internet connection in a place where they felt safe discussing issues related to sexual healthcare and HIV prevention. Participants were recruited through flyers posted in locations around the Birmingham, Alabama metropolitan area, including local clinics and community organizations serving transgender individuals, bars, clubs, breweries, and various locations across the University of Alabama at Birmingham (UAB) campus. Participants were also recruited by clinicians and

other clinic staff at the UAB Gender Health Clinic (GHC). Community stakeholders and transgender community organizations also posted advertisements on their social media channels (i.e., Facebook, Instagram, Twitter) and advertised this study through word of mouth. Snowball sampling was also utilized; participants were given business cards with study details to hand out to friends after they completed the study. Participants received a $50 Visa gift card for their participation in the study. If participants referred someone to the study and that person enrolled, the referring participant received an additional $20 in compensation for each referral up to four times, for a total of $80 of additional reimbursement.

This study was approved by the University of Alabama at Birmingham (UAB) Institutional Review Board (IRB) (protocol #IRB-300005085). Participants provided verbal consent as the focus groups were conducted virtually amidst the COVID-19 pandemic. A copy of the consent form was emailed to each participant for their review. A member of the study staff also spoke with participants over the phone to review the informed consent document after participants had a chance to review it and answer any questions. Participants either e-signed the consent form and emailed it back to study staff (if they were able) or verbally consented with study staff, who documented this on a printed version of the consent form. This consent process was approved by the UAB IRB and was completed prior to participation in any study related activities.

## Virtual focus groups

Participants were called by study staff and screened for eligibility after inquiring about the study through the recruitment methods mentioned above. All participants underwent verbal informed consent over the phone and agreed to all study procedures, including participation in the VFG platform and permission for video and audio recording of the VFG. They were also emailed a copy of the consent form. Once deemed eligible and informed consent was obtained, participants remained on the phone and completed an interviewer-administered questionnaire on their socio-demographics (e.g., age, race/ethnicity, insurance status), sexual history, STI history, details regarding gender transition, and history of PrEP use. After completing the questionnaire participants provided an email address to study staff, were sent a link to a password protected Zoom meeting, and were informed of the time and date of their 60-minute VFG session. All VFGs were facilitated by the same moderator (author E. L.A), who has extensive experience in qualitative research involving sexual and gender minorities [23–25]. Each VFG was audio and video recorded via the Zoom platform. The audio file of the session was stored in a secure, online cloud and later sent to a professional transcription service.

## Theoretical framework: The health belief model

A standardized discussion guide for VFG sessions was developed based on theoretical constructs of the health belief model (HBM), which include perceived barriers, benefits, susceptibility, severity, and potential cues to action as they relate to a particular disease or situation (HIV and HIV PrEP, in this study) [26]. VFG moderator questions and how they are linked to theoretical constructs in the HBM can be found in Table 1. The HBM framework was selected to guide the VFG discussions in order to help our team understand how TGW in the Southeastern U.S. view HIV risk and the appropriateness of PrEP for their community. This is a critical bridge between participants' lived experiences and providers' desire to provide them with an option to protect themselves from HIV. This theoretical construct has been used to study HIV prevention and sexual risk behaviors in multiple populations, including black women and adolescents [27–29].

**Table 1. Virtual focus group questions derived from theoretical constructs.**

| Health Belief Model construct | Focus group prompt questions |
|---|---|
| Perceived susceptibility | • Tell me about HIV in this community. Do you feel like that's currently a major concern for people you know?<br>• Do most people in this community feel like they're at risk for HIV infection? |
| Perceived severity | • Living with HIV has changed so much over time. How do people in this community feel about HIV these days? |
| Perceived benefits | • In recent years PrEP has become widely available. What do you think about that– what do you see as the benefits of PrEP for people in this community? |
| Perceived barriers | • Why do you think use of PrEP is still pretty low among people in this community– what are the barriers? |
| Cues to action | • What would make you want to look into being prescribed PrEP? Hearing from other people in this community? Hearing from health care providers? Seeing ads for PrEP? |

## Qualitative analysis

Because this study was built around an existing theoretical framework, thematic coding was used for the qualitative data analysis [30, 31]. Familiarity with the data was established from the outset, with multiple team members (authors O.T.V.G., A.F.C., L.V.O., C.A.M.) observing the focus groups as they were conducted [32]. Transcripts were reviewed and discussed in team meetings as soon as they became available, with initial codes and emerging themes noted in debriefing memos. The focus group guide was modified in an iterative process after each team discussion to reflect emerging themes. During the final focus group, member checking (i.e., asking participants to provide feedback on our interpretations of the data) was used to establish the validity of the findings [33].

NVivo qualitative data analysis software was used to organize the thematic coding [34]. Initial codes were derived from the theoretical model (e.g., susceptibility, barriers); as analysis continued, additional subcodes were developed to provide a more refined view of the data. Once all codes and subcodes were established, two authors (O.T.V.G. and E.L.A.) coded each transcript independently to explore the reliability of the analysis and there were no disagreements in coding between them. Following a detailed examination of patterns in the coding, several subcodes were merged or eliminated based on the emergent understanding of the data. Based on repeated readings of the transcripts, these same authors developed the final themes and identified illustrative quotes to ensure the confirmability of the results [32].

## Results

### Sample characteristics

Between July 2020 and December 2020, 27 TGW inquired about participation in the study and were screened via phone. Details of screening and enrollment are shown in Fig 1. Four TGW were excluded based on self-reported HIV-positive status and 1 was excluded for being under 18 years old. Two TGW who were initially interested declined to participate after calling to complete the survey and consent process. One of these women cited that she did not feel she would be helpful in the study objective since she was not sexually active and the other felt uncomfortable with the subject matter surrounding PrEP, expressing that she felt it was targeting TGW as high risk. Two TGW were screened for eligibility during visits at the UAB GHC and completed the questionnaire and consent process at that time but were unable to be contacted for the VFG. One additional TGW completed the screening, questionnaire, and consent

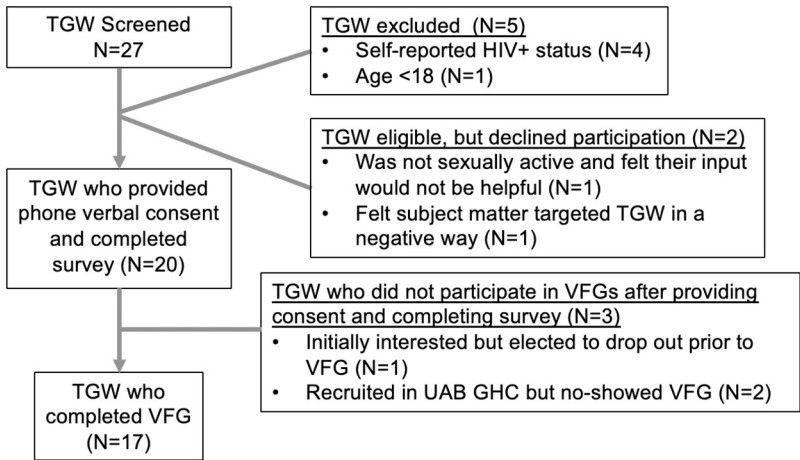

**Fig 1. Flow diagram of screening and enrollment of transgender women for virtual focus groups.**

process, however no showed several VFG sessions. She was contacted prior to the final VFG and said she was no longer interested in participating but did not cite a reason.

Thus, seventeen eligible TGW enrolled in the study and each of them participated in one of four VFGs. Characteristics of these study participants are shown in Table 2. Mean participant age was 28.1±8.5 years. Black, non-Hispanic and white, non-Hispanic TGW represented 41% (n = 7) and 47% (n = 8) of participants, respectively. Participants reported a variety of sexual orientations with 35.3% identifying as heterosexual (n = 6), 23.4% as homosexual (n = 4), 17.6% as bisexual (n = 3), 11.8% as pansexual (n = 2), 5.9% as fluid (n = 1), and 5.9% saying they did not know (n = 1). They also reported the gender(s) of their lifetime sexual partners, with 82.3% reporting sex with cisgender men (n = 14). Regarding number of lifetime sexual partners, participants reported a range of 0 to "5 million." Most participants (53.9%, n = 9) reported between 0–5 lifetime sexual partners. Only two focus group participants (11.8%) reported having had an STI in their lifetime. Fifteen participants had heard of PrEP but only one had ever taken it for HIV prevention. All participants resided in Alabama, with many citing experiences living in rural parts of the state either at the time of the focus groups or in the past.

Focus group findings (themes and representative quotes detailed in Table 3).

**Perceived susceptibility.** Each VFG began with an exploration of perceptions of susceptibility to HIV infection among TGW, as perceiving oneself to be at risk for a particular negative health outcome is a necessary precursor to adopting protective health behaviors such as PrEP use. TGW in each VFG repeatedly emphasized that the risk for HIV infection is a function of an individual's risk behaviors (including commercial sex work, multiple sexual partners, and condomless anal intercourse), *not* their identity. While participants noted that some TGW do engage in high-risk sexual behaviors, they emphasized the numerous pathways through which pervasive discrimination against transgender individuals (i.e., commercial sex work necessitated by poverty, vulnerability to violent sexual assault and intimate partner violence, untreated mental health issues, etc.) may lead to behaviors that increase susceptibility to HIV infection. Participants pushed back strongly, however, against the supposition that TGW as a group are equally susceptible to HIV infection; rather, they urged that both individual behaviors and social circumstances be considered when addressing the suitability of HIV prevention measures such as PrEP for this population.

**Table 2.  Characteristics of transgender women participating in focus groups (n = 17).**

|  | N (%) or Mean ± SD |
| --- | --- |
| **Age** | 28.1 ± 8.5 |
| **Race/ethnicity** |  |
| Black, non-Hispanic | 7 (41.1%) |
| White, Hispanic | 1 (5.9%) |
| White, non-Hispanic | 8 (47.0%) |
| White, unknown ethnicity | 1 (5.9%) |
| **Educational level** |  |
| High school/GED | 5 (29.4%) |
| Some college/associate's degree | 9 (52.9%) |
| Bachelor's degree | 1 (5.9%) |
| Any post-graduate studies | 2 (11.8%) |
| **Sexual orientation** |  |
| Bisexual | 3 (17.6%) |
| Heterosexual | 6 (35.3%) |
| Homosexual | 4 (23.5%) |
| Fluid | 1 (5.9%) |
| Pansexual | 2 (11.8%) |
| Don't know | 1 (5.9%) |
| **Gender of sexual partners*** |  |
| Transgender women | 4 (23.5%) |
| Transgender men | 3 (17.6%) |
| Cisgender women | 6 (35.5%) |
| Cisgender men | 14 (82.3%) |
| Genderfluid individuals | 3 (17.6%) |
| **Number of lifetime sexual partners** |  |
| 0–5 | 9 (53.9%) |
| 6–20 | 4 (23.5%) |
| 21–50 | 0 (0.0%) |
| 51–100 | 2 (11.8%) |
| >100 | 2 (11.8%) |
| **Self-reported STI history** | 2 (11.8%) |
| **Participation in transactional sex** | 5 (29.4%) |
| **Currently using HRT** | 13 (76.5%) |

*Participants reported multiple genders of their sexual partners.

Abbreviations: HRT = hormone replacement therapy; SD = standard deviation.

In addition to the distinction between risky behaviors and identity, participants also noted how assumptions about TGW's HIV risk derive from the historical and ongoing association between HIV/AIDS and the Lesbian, Gay, Bisexual, Transgender, Queer Plus (LGBTQ+) community. Several participants shared that, although they identify as lesbian or are not sexually active at all, medical providers and society in general portray them as susceptible to HIV infection simply due to their inclusion in the LGBTQ+ community. They expressed that the conflation of HIV risk with transgender identify is insulting and perpetuates commonly held misconceptions of the complex reality of the transgender experience.

**Perceived severity.**    While TGW in this study acknowledged that HIV was a concern for individuals who participate in high-risk sexual behaviors, the perceived severity of HIV

**Table 3. Themes and representative quotes from transgender women, stratified by health belief model theoretical constructs.**

| Perceived Susceptibility | |
|---|---|
| • Frustration with conflation of HIV risk with transgender identity<br>• HIV and PrEP are not for transgender women, but rather for cisgender gay men | *"People may judge them for taking it because like stereotypically like. . . like PrEP is stereotypically meant for men who sleep with other men. So like trans women may feel as if they're not meant to take it because they're women, you know."*<br>*". . .they always refer to our lives as risky behaviors, why do we have to be referred to as risky behaviors? That's so. . . I mean it's just. . . it's crazy. . . my life is not a risky behavior and none of my trans sisters are risky behavior, either. . ."*<br>*"A lot of cis people just don't get it, you know. . .I've had a lot of people try but they just do not get the fact that just because you're trans. . .you're not out there partying, you're not, you know, a sex worker, you know, you don't do hard drugs just because you're trans. . ."* |
| **Perceived Severity** | |
| • Low perceived susceptibility of HIV infection<br>• Heightened salience of trans-related health issues (i.e., hormone therapy and other transition related therapies, mental health) | *"Well, me personally, I feel like I already know what the best option for me is going to be because the goal for me is going to be because the goal for me is to become. . . become my authentic self. And I don't need anything that is going to, you know, distract me or, you know, not let me get to my authentic self. So I don't need any blocks in the road so I'm pretty sure what I'm gonna do in the long run"*<br>*"The focus in all the trans communities that I've ever been a part of has strayed from physical health and much into mental health. If we're talking about our health it's generally like mentally coping essentially"* |
| **Benefits** | |
| • Few benefits identified in the setting of limited knowledge of PrEP<br>• Viewed as a general sexual health promotion tool | *"I didn't even know apparently there's a drug that stops HIV now that they give to some of the trans women sometimes. . ."*<br>*"I really don't know much about it; I just know that if you're going to be sexually active you should probably get on it, and that's literally all I know."*<br>*"For me, I've heard a lot of good things about it, so I just imagine they're just being really responsible with their body and they're doing what they're doing."*<br>*"So for me, once again, it's not really a topic, but some of my trans friends that I know, um, they actually are already on PrEP, so like they love PrEP and they're always telling me I need to get on PrEP. . ."* |
| **Barriers** | |
| • Limited resources (i.e., lack of transportation, affordability)<br>• Inappropriate or inaccessible trans-centered healthcare<br>• Drug-drug interactions and potential side effects<br>• Limited knowledge about PrEP in general | *"And one thing that I've seen help individuals as far as to be crystal clear with what it does to the body, the medication itself, because usually when you are talking, when you learn about hormones you usually have to go through every single step and you learn what it does to the body, how it interacts with it and what changes would happen. And usually with that kind of understanding if they're presented with a new drug as long as it's defined well enough and clear enough to the individual they're more willing to accept it or more willing to realize. . .how it will actually affect them instead of speculating into the unknown and more about worry about potential harm."*<br>*"When it comes to PrEP I don't think trans women were thought about when the drug was created, that's my personal opinion. The reason I say that is because it has such a negative effect on us as far as it relates to us and our HRT."*<br>*"And then you also have to think about it, there's always this thought in the back of your head, we're taking hormones, what are the side effects that it will have when you're on hormones and the PrEP."*<br>*"I personally decided not to take PrEP because of the negative effects that it would have on my HRT, so I did not take it."* |
| **Cues to Action** | |
| • Inappropriate representation of transgender women in media | *"And even in today's commercials for PrEP products or HIV commercials they just started adding trans women and then if you really look at it some of 'em are just drag queens instead of actual trans women."*<br>*". . .people, you know, want to see representation of themselves. So, you know, if you put a trans woman or black trans woman on . . . PrEP commercials trans women will feel more prone to believe, you know, that it's applicable for us. I think that only recently they had one PrEP commercial with Hayley Sahara from Pose. I think that just was a recent commercial. As a trans woman, she was depicted in a PrEP commercial and that's the only one I ever seen. So only now are we beginning, you know, just splashing the surface of being able to create representation centered around trans women, you know, for PrEP and HIV, and not just grouping us in with men sleeping with men or gay men."*<br>*"Like for me, I love Janet Mock and Laverne Cox and even though they're not really Instagram influencers I would love to see them talk about it because I trust them."*<br>*"It needs to be someone that's covered the trans community that when they speak up for the trans community they're speaking up wholeheartedly; they're just not goin' off something they were advocating on."* |

infection as a health outcome was relatively low for many participants. They reported greater importance of both physical and mental health among TGW as an overarching theme throughout the focus groups; several participants explicitly stated that successfully transitioning was the single most important consideration. Other concerns, including HIV prevention and even HIV infection, were secondary to the central goal of living as their authentic selves.

**Benefits.**   Overall, TGW in the focus groups could name few concrete benefits associated with the use of PrEP to prevent HIV infection. This appeared to result from participants' limited consideration of PrEP, rather than a careful weighing of the pros and cons of PrEP use. While only one participant reported absolutely no knowledge of PrEP prior to recruitment in this study, most participants generally associated PrEP use with safer sex practices.

**Barriers.**   Participants in the study noted many barriers to PrEP use typically experienced by marginalized and/or under-resourced populations, including lack of transportation to clinics offering PrEP and concerns about the affordability of PrEP due to lack of health insurance. Participants noted additional concerns about inappropriate or inaccessible healthcare that are unique to the transgender population, including a perceived lack of trans-specific expertise on the part of health care providers. Participants reported concerns about potential interactions between PrEP medications and the use of gender-affirming hormone therapy, including both unexpected side effects and, more critically, the possibility that the effects of hormone therapy would be diminished by PrEP. Participants acknowledged their own limited information on PrEP use (which they attributed both to poor sex education and coming from rural areas where PrEP was not available) but expressed concern that their medical providers also lacked critical information on how PrEP might affect TGW in unexpected ways.

**Cues to action.**   The decision to adopt new or innovative health behaviors can be powerfully shaped by the internal and external cues to action individuals receive [26]. Participants noted numerous external cues to action related to PrEP use, but found most to be highly problematic. The most animated discussions during the focus group sessions centered around depictions of TGW in television commercials for PrEP, which participants felt were not an accurate representation of real TGW and therefore not relevant to their lives. In discussing PrEP commercials, participants also reiterated their view that PrEP was developed for and remains primarily targeted to MSM, leaving TGW's unique concerns about PrEP completely out of the conversation.

While current TV commercials were largely ineffective as cues to action, participants spoke positively of the various social media channels such as YouTube and Instagram where they follow well-known TGW. Due to their authenticity, these influencers were viewed as reliable sources of information for other TGW; participants expressed an openness to hearing about the benefits of PrEP for TGW from these sources. Participants further explained their reliance on social media for accurate information by noting that the local transgender community does not actively engage in messaging around PrEP use.

## Discussion

To our knowledge, this is the first qualitative study to engage TGW in Alabama to assess their attitudes about HIV prevention and PrEP. The virtual format allowed participants from all over the state, including many currently or previously residing in rural areas, to share their perspectives. While sexual healthcare resources, including PrEP, are sparse in the rural Southeastern U.S., many LGBTQ+ people live in these areas [12]. Sexual health research inclusive of rural-dwelling individuals is extremely limited, thus capturing the insights of rural TGW in Alabama is novel and valuable since these women face unique struggles in terms of access and sexual health education.

Overall, we found that most TGW in our sample were aware of PrEP's existence and value as an HIV prevention tool, but only one participant had used it. Limited PrEP uptake in the setting of adequate awareness stems from several different vantage points. In the context of the HBM, perceived susceptibility to HIV risk was variable given the diversity of sexual behaviors represented in this sample of TGW. For those who did not perceive their risk of HIV to be high, this may be partially due to the overall shift in perceptions of HIV since the introduction of highly effective antiretroviral therapy in the 1990s [35] and the resulting reframing of HIV infection as a chronic rather than terminal condition. For these participants, however, views on the perceived severity of HIV appeared to be a function of the heightened salience of trans-related health issues.

In addition, many participants simply felt that PrEP was not meant for them—either because they saw it as a sexual health promotion tool designed for cisgender MSM or because they did not engage in any HIV risk behaviors that would warrant PrEP. Their perceived risk being relatively low was actually in keeping with their self-reported number of lifetime sexual partners and STI history, with the majority of participants reporting 0–5 lifetime sexual partners and/or no history of STIs. This underscores the importance of avoiding assumptions when providers are discussing sexual health with transgender people—this population represents a wide variety of sexual orientations, practices, and genders of sexual partners. Assumptions that TGW engage in sexual risk behaviors could greatly harm the patient-provider relationship by perpetuating stereotypes of this population and mistrust of the medical establishment.

The perceived severity of HIV infection was blunted by competing priorities identified by our sample, most importantly gender-affirming hormone therapy. It was clear that any medications that may interfere with one's transition were deemed not worthwhile by TGW in this study. Some even conveyed concerns that HIV PrEP would interfere with their HRT, citing this as major barrier to taking it. Data are clear that PrEP does not impact the levels of feminizing hormone levels [36, 37], but participants were still very concerned about the potential for this. Based on the experiences of our sample, there are limited healthcare providers or PrEP advocates in the Southeastern U.S. disseminating transgender-specific, evidence-based information about PrEP. This disparity underscores the need for further engagement of TGW in PrEP research as well as enhanced educational efforts centered on this population.

Future studies are needed to pilot community-driven, patient-centered solutions (e.g., at-home STI testing options, telePrEP services) to sexual health promotion and HIV prevention in this population. Intervention development with the input of TGW living in the Southeast, who can share their lived experiences and expound upon the result of this study, is essential. Regarding broader societal barriers impacting HIV prevention efforts for this population (e.g., stigma, discrimination), persistent advocacy and public policy influence are needed to promote an accepting and affirming culture in the Southeast.

## Limitations

Results from this study are not generalizable to all TGW due to the study only taking place in Alabama, the recruitment strategies used, and the resulting sample of participants. Although we recruited participants for this study through various methods, all TGW in our study were connected enough to transgender resources locally to learn about the study and in stable enough living situations to have internet access. By their own admission, participants in this study were not representative of TGW whose marginalization may result in high-risk behaviors as they cited limited experiences with commercial sex work and condomless sex with multiple partners. Regardless of these limitations, the TGW in this study represent a portion of the

larger transgender population whose sexual health care needs must also be carefully considered.

This study was also conducted during the COVID-19 pandemic at a time when in person focus groups were not possible. As such, assessment of group dynamics and body language were limited on the Zoom platform since participants were able to turn their video feature off. We did, however, find that conducting VFGs via Zoom allowed greater ease of participation for TGW who did not live in the Birmingham metropolitan area (i.e., those living in rural locations in Alabama) and for those who did not feel comfortable coming to the university campus for a group session for safety reasons. While access to safe and reliable internet connections may be a limitation for some, virtual platforms for qualitative research may be useful in engaging this population in future studies.

## Conclusions

Nuanced messaging harnessing the unique needs and characteristics of the transgender community in the Southeastern U.S. is necessary to properly educate and engage TGW in HIV prevention strategies such as PrEP. A one-size-fits-all approach is inappropriate given the diversity among TGW regarding sexual behaviors and HIV risk behaviors. Discussions between TGW and healthcare providers should focus on individual HIV risk and patient concerns when determining whether PrEP is appropriate.

## Supporting information

**S1 File.**
(DOCX)

**S2 File.**
(DOCX)

**S3 File.**
(DOCX)

**S4 File.**
(DOCX)

## Acknowledgments

The authors would like to thank the transgender women of Alabama for sharing their experiences with our research team. We would also like to acknowledge Brianna Patterson, Birmingham AIDS Outreach, Magic City Wellness, and the Birmingham Bevy for their partnership and help with recruitment for this study. Drs. Latesha Elopre and Susan Davies were also instrumental in providing guidance in the development of the focus group script and community-based recruitment strategies. This project was presented as an oral presentation at the 27th Annual Agency for Health Research and Quality National Research Service Award Trainees Research Conference (Virtual) on June 8, 2021 as well as the 2021 STI & HIV World Congress (Virtual) on July 13, 2021 (oral presentation #54). It was also presented as a poster presentation at the 2021 UAB Department of Medicine Trainee Research Symposium, Birmingham, AL on March 3, 2021, where it was selected as a semi-finalist for the Joseph Reeves Award for Excellent in Research by a Post-Doctoral Scholar.

## Author Contributions

**Conceptualization:** Olivia T. Van Gerwen, Christina A. Muzny.

**Data curation:** Olivia T. Van Gerwen, Erika L. Austin, Andres F. Camino, L. Victoria Odom.

**Formal analysis:** Erika L. Austin.

**Funding acquisition:** Olivia T. Van Gerwen, Christina A. Muzny.

**Investigation:** Andres F. Camino, L. Victoria Odom.

**Methodology:** Erika L. Austin.

**Project administration:** Olivia T. Van Gerwen.

**Resources:** Erika L. Austin.

**Visualization:** Olivia T. Van Gerwen, Christina A. Muzny.

**Writing – original draft:** Olivia T. Van Gerwen.

**Writing – review & editing:** Olivia T. Van Gerwen, Erika L. Austin, Andres F. Camino, L. Victoria Odom, Christina A. Muzny.

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
