## [Decision Letter · Decision Letter 0]

23 Nov 2021

PONE-D-21-33143"It's behaviors, not identity": Attitudes and Beliefs Related to HIV Risk and Pre-Exposure Prophylaxis Among Transgender Women in the Southeastern United StatesPLOS ONE

Dear Dr. Olivia T. Van Gerwen,

Thank you for submitting your manuscript to PLOS ONE. After careful consideration, we feel that it has merit but does not fully meet PLOS ONE’s publication criteria as it currently stands. Therefore, we invite you to submit a revised version of the manuscript that addresses the points raised during the review process.

We look forward to receiving your revised manuscript.

Kind regards,

Kimani Makobu

Academic Editor

PLOS ONE

Journal Requirements:

a) Did participants provide their written or verbal informed consent to participate in this study?

 [This work was supported by the UAB Center for AIDS Research [P30 AI027767-32 to OTVG] and the Agency of Healthcare Research and Quality [T32HS013852 to OTVG].]

[I have read the journal's policy and the authors of this manuscript have the following competing interests: OTVG has received research grant support from Gilead Sciences, Inc. and Abbott Molecular. CAM has received research grant support from NIH/NIAID, Gilead Sciences, Inc., and Lupin Pharmaceuticals, Inc.; serves as a consultant for Abbott Molecular, Lupin Pharmaceuticals, Inc., Cepheid, and BioFire Diagnostics; receives honoraria from Elsevier, Abbott Molecular, Cepheid, Becton Dickinson, Roche Diagnostics, and Lupin Pharmaceuticals, Inc. All other authors have no relevant disclosures.]

6. Please include your tables as part of your main manuscript and remove the individual files. Please note that supplementary tables (should remain/ be uploaded) as separate "supporting information" files.

Additional Editor Comments:

Well written manuscript addressing gaps in PrEP use among TGW in Southeastern US

1. Background:

• The authors state that PrEP uptake for TGW is low. It would be helpful to present data for PrEP uptake among TGW compared to that of other population types. If there are data available, please indicate estimates of TGW at risk and the proportion of these that are on PrEP.

2. Methods

• May be useful to indicate and cite instances where the health belief model has been used in similar or related work

• Were there disagreements in coding? Please explain how they were resolved.

3. Results

• Lines 216-219 – this statement appears to be an interpretation of results. You may consider moving this to Discussion

4. Discussion:

• Line 283-284 – participants did not feel PrEP was for them because they did not engage in HIV risk behavior. Did the authors explore whether this perception was correct either through the qualitative work or by examining data obtained in the questionnaires e.g., what was the prevalence of STIs, # of sex partners etc.

• The authors have detailed a couple of barriers to PrEP use. It may be useful to discuss whether such barriers can be addressed as way of next steps following on from this work.

Other:

Line 49: is the last ‘through’?

Reviewers' comments:

Reviewer's Responses to Questions

**Comments to the Author**

1. Is the manuscript technically sound, and do the data support the conclusions?

Reviewer #1: Yes

2. Has the statistical analysis been performed appropriately and rigorously? 

Reviewer #1: N/A

3. Have the authors made all data underlying the findings in their manuscript fully available?

Reviewer #1: Yes

4. Is the manuscript presented in an intelligible fashion and written in standard English?

Reviewer #1: Yes

5. Review Comments to the Author

Reviewer #1: Well written manuscript addressing gaps in PrEP use among TGW in Southeastern US

1. Background:

• The authors state that PrEP uptake for TGW is low. It would be helpful to present data for PrEP uptake among TGW compared to that of other population types. If there are data available, please indicate estimates of TGW at risk and the proportion of these that are on PrEP.

2. Methods

• May be useful to indicate and cite instances where the health belief model has been used in similar or related work

• Were there disagreements in coding? Please explain how they were resolved.

3. Results

• Lines 216-219 – this statement appears to be an interpretation of results. You may consider moving this to Discussion

4. Discussion:

• Line 283-284 – participants did not feel PrEP was for them because they did not engage in HIV risk behavior. Did the authors explore whether this perception was correct either through the qualitative work or by examining data obtained in the questionnaires e.g., what was the prevalence of STIs, # of sex partners etc.

• The authors have detailed a couple of barriers to PrEP use. It may be useful to discuss whether such barriers can be addressed as way of next steps following on from this work.

Other:

Line 49: is the last ‘through’?

6. PLOS authors have the option to publish the peer review history of their article (what does this mean?). If published, this will include your full peer review and any attached files.

Reviewer #1: No

---

## [Author Response · Author response to Decision Letter 0]

9 Dec 2021

Dear Journal Editor(s), 

Thank you to the reviewers for their comments on our manuscript entitled “’It's behaviors, not identity’: Attitudes and Beliefs Related to HIV Risk and Pre-Exposure Prophylaxis Among Transgender Women in the Southeastern United States.” Below you will find an itemized list of our responses to each of the reviewers’ comments. A revised manuscript with tracked changes in addition to a clean copy have been uploaded to the manuscript submission portal. All line numbers responding to reviewer comments refer to the tracked version of the manuscript

Journal Requirements:

Author Response: We have updated the manuscript to follow PLOS ONE’s style requirements.

a) Did participants provide their written or verbal informed consent to participate in this study?

Author Response: This section has been updated to provide details on the consent process for this study and added to the revised Methods Section (Page 5, Lines 156-169)

3. Thank you for stating the following financial disclosure: [This work was supported by the UAB Center for AIDS Research [P30 AI027767-32 to OTVG] and the Agency of Healthcare Research and Quality [T32HS013852 to OTVG].] Please state what role the funders took in the study. If the funders had no role, please state: "The funders had no role in study design, data collection and analysis, decision to publish, or preparation of the manuscript." If this statement is not correct you must amend it as needed. Please include this amended Role of Funder statement in your cover letter; we will change the online submission form on your behalf.

Author Response: The financial disclosure/funding statement has not changed since the initial submission. The funders had no role in study design, data collection and analysis, decision to publish, or preparation of the manuscript. 

4. Thank you for stating the following in the Competing Interests section: [I have read the journal's policy and the authors of this manuscript have the following competing interests: OTVG has received research grant support from Gilead Sciences, Inc. and Abbott Molecular. CAM has received research grant support from NIH/NIAID, Gilead Sciences, Inc., Abbott Molecular, and Lupin Pharmaceuticals, Inc.; serves as a consultant for Abbott Molecular, Lupin Pharmaceuticals, Inc., Cepheid, and BioFire Diagnostics; receives honoraria from Elsevier, Abbott Molecular, Cepheid, Becton Dickinson, Roche Diagnostics, and Lupin Pharmaceuticals, Inc. All other authors have no relevant disclosures.] Please confirm that this does not alter your adherence to all PLOS ONE policies on sharing data and materials, by including the following statement: "This does not alter our adherence to PLOS ONE policies on sharing data and materials.” (as detailed online in our guide for authors http://journals.plos.org/plosone/s/competing-interests). If there are restrictions on sharing of data and/or materials, please state these. Please note that we cannot proceed with consideration of your article until this information has been declared. 

Author Response: This does not alter our adherence to PLOS ONE policies on sharing data and materials.

Author Response: The above Competing Interests section has been revised to state that first author OTVG and senior author CAM has also received research grant support from Abbott Molecular.

Author Response: The ethics statement has been moved to the Methods section in the revised manuscript (Page 5, Lines 156-169)

7. Please include your tables as part of your main manuscript and remove the individual files. Please note that supplementary tables (should remain/ be uploaded) as separate "supporting information" files.

Author Response: The tables (Tables 1-3) have been integrated into the main manuscript. We did not have any supplemental tables for this submission.

8. Please review your reference list to ensure that it is complete and correct. If you have cited papers that have been retracted, please include the rationale for doing so in the manuscript text or remove these references and replace them with relevant current references. Any changes to the reference list should be mentioned in the rebuttal letter that accompanies your revised manuscript. If you need to cite a retracted article, indicate the article’s retracted status in the References list and also include a citation and full reference for the retraction notice.

Author Response: All references have been reviewed and meet criteria to be included in this manuscript and are correctly formatted. None have been retracted.

Additional Editor Comments:

1. The authors state that PrEP uptake for TGW is low. It would be helpful to present data for PrEP uptake among TGW compared to that of other population types. If there are data available, please indicate estimates of TGW at risk and the proportion of these that are on PrEP.

Author Response: As we note in the introduction of the manuscript, data are quite limited regarding PrEP uptake among TGW and much of the data that do exist in the literature aggregate TGW with cisgender MSM. We have re-visited the literature and added data from more recent studies citing low PrEP uptake among TGW and compared these data to that published about cisgender MSM. These data also describe how many TGW in these studies were eligible for PrEP and the proportion who were taking it. Unfortunately, there are no studies that we could find that directly compare MSM and TGW in terms of PrEP uptake in the same study, so we added two sentences describing what is known in this space currently: “One recent multi-site U.S. cohort study of 600 TGW identified 47% to be eligible for PrEP, but only 28% of those reported using PrEP in the last 30 days. Use of PrEP among eligible MSM, by comparison, was estimated to be 35% as of 2017 in another U.S. study.” (Page 3, Lines 107-111). 

2. Methods: May be useful to indicate and cite instances where the health belief model has been used in similar or related work.

Author Response: The Health Belief Model has been used extensively in sexual risk behavior and HIV prevention research. We have added a statement along these lines and added 3 references of such previous work. (Page 7, Lines 200-202, References 27-29)

3. Methods: Were there disagreements in coding? Please explain how they were resolved.

Author Response: There were not any disagreements in coding, which we have explained in the Methods section of the revised manuscript. (Page 8, Line 222)

4. Results: Lines 216-219 – this statement appears to be an interpretation of results. You may consider moving this to Discussion

Author Response: This statement was removed from the Results and added to the Discussion section of the revised manuscript. (Page 17, Line 371-376)

5. Discussion: Line 283-284 – participants did not feel PrEP was for them because they did not engage in HIV risk behavior. Did the authors explore whether this perception was correct either through the qualitative work or by examining data obtained in the questionnaires e.g., what was the prevalence of STIs, # of sex partners etc.

Author Response: Thank you for this interesting, important comment. Our demographic survey asked participants to self-report STI history and disclose number of lifetime sexual partners, so we have added these data to Table 2 (Pages 9-10). We have also added a sentence to the results section stating that the majority of our participants (>50%) reported 0-5 lifetime sexual partners and only 11.8% of participants reported an STI in their lifetime. (Page 9, Lines 253-255) This was in keeping with the perceived low HIV risk reported by our group of participants, so we added a sentence saying this to the Discussion (Page 17, Lines 379-382)

6. Discussion: The authors have detailed a couple of barriers to PrEP use. It may be useful to discuss whether such barriers can be addressed as way of next steps following on from this work.

Author Response: We have added a section to the discussion addressing this important point. We feel that next steps that could help break down the identified barriers would be using the qualitative data from this study to develop sexual health promotion and HIV prevention efforts tailored to the TGW community of the Southeastern US. Some things that may be considered include at-home STI test collection and telePrEP. Of course, some of the larger barriers noted by participants (e.g., stigma, discrimination) are more difficult to target and will require high-level advocacy and public policy influence, which we also briefly discuss. (Page 18, Line 403-409)

7. Other: Line 49: is the last ‘through’?

Author Response: Correct. This typo has been corrected. (Now on Page 2, Line 64)

Please do not hesitate to contact us should you have any questions regarding our responses to the reviewers’ comments or if you require any additional revisions to our manuscript.

---

## [Editor Report · Decision Letter 1]

20 Dec 2021

"It's behaviors, not identity": Attitudes and Beliefs Related to HIV Risk and Pre-Exposure Prophylaxis Among Transgender Women in the Southeastern United States

PONE-D-21-33143R1

Dear Dr. Van Gerwen,

We’re pleased to inform you that your manuscript has been judged scientifically suitable for publication and will be formally accepted for publication once it meets all outstanding technical requirements.

Kind regards,

Kimani Makobu

Academic Editor

PLOS ONE

Additional Editor Comments (optional):

Just for information, while data comparing PrEP uptake in transgender women and cis gender MSM are limited there is a publication from East Africa that seems to have done this in a small cohort. Please consider citing them.

https://pubmed.ncbi.nlm.nih.gov/33465090
---

## [Editor Report · Acceptance letter]

13 Jan 2022

PONE-D-21-33143R1 

“It’s behaviors, not identity”: Attitudes and beliefs related to HIV risk and pre-exposure prophylaxis among transgender women in the Southeastern United States 

Dear Dr. Van Gerwen:

I'm pleased to inform you that your manuscript has been deemed suitable for publication in PLOS ONE. Congratulations! Your manuscript is now with our production department. 

Kind regards, 

on behalf of

Dr. Kimani Makobu 

Academic Editor

PLOS ONE